# Strip cropping shows promising increases in ground beetle community diversity compared to monocultures

Luuk Croijmans[1,2]*[†], Fogelina Cuperus[2,3][†], Dirk F van Apeldoorn[2,3], Felix JJA Bianchi[2]*, Walter AH Rossing[2], Erik H Poelman[1]

[1]Laboratory of Entomology, Wageningen University and Research, Wageningen, Netherlands; [2]Farming Systems Ecology, Wageningen University and Research, Wageningen, Netherlands; [3]Field Crops, Wageningen University and Research, Lelystad, Netherlands

*For correspondence:
luuk.croijmans@wur.nl (LC);
felix.bianchi@wur.nl (FJJAB)

[†]These authors contributed
equally to this work

Competing interest: The authors
declare that no competing
interests exist.

Reviewing Editor: Bernhard
Schmid, University of Zurich,
Switzerland

## eLife Assessment

This study presents **important** findings on increased ground beetle diversity in strip cropping compared with crop monocultures. **Solid** methods are used to analyze data from multiple sites with heterogeneous systems of mixed crops, allowing broad conclusions, albeit at the expense of lacking taxonomic specificity. The work will be of interest to all those applying plant diversity treatments to improve the diversity of associated animals in agricultural fields.

**Abstract** Global biodiversity is declining at an unprecedented rate, with agriculture as one of the major drivers. There is mounting evidence that intercropping can increase insect biodiversity while maintaining or increasing yield. Yet, intercropping is often considered impractical for mechanized farming systems. Strip cropping is a type of intercropping that is compatible with standard farm machinery and has been pioneered by Dutch farmers since 2014. Here, we present ground beetle data from four organically managed experimental farms across four years. Ground beetles are sensitive to changes in habitats and disturbances, and hold keystone positions in agroecosystem food webs. We show that strip cropping systems can enhance ground beetle biodiversity, while other studies showed that these increases have been achieved without incurring major yield loss. Strip-cropped fields had on average 15% more ground beetle species and 30% more individuals than monocultural fields. The higher ground beetle richness in strip crops was explained by the merger of crop-related ground beetle communities, rather than by ground beetle species unique to strip cropping systems. The increase in field-level beetle species richness in organic agriculture through strip cropping approached increases found for other readily deployed biodiversity conservation methods, like shifting from conventional to organic agriculture (+19% −+23%). This indicates that strip cropping is a potentially useful tool supporting ground beetle biodiversity in agricultural fields without compromising food production.

## Introduction

Insects account for 80% of the animal species in the world and, therefore, the recently reported unprecedented rate of insect decline is cause for alarm about the state of biodiversity on Earth (*Dirzo et al., 2014*; *Hallmann et al., 2017*; *van Klink et al., 2020*). An array of biodiversity metrics provides strong evidence for declines of especially terrestrial insects across all continents (*van Klink et al., 2020*). This includes declines in abundance of both common and rare species (*Hallmann et al., 2017*;

**eLife digest** Insects are the largest and most diverse group of animals, comprising approximately 80% of all animal species. They inhabit nearly every place on Earth and play important and varied roles in ecosystems, from serving as a food source for other animals and as recyclers of organic matter and nutrients, to acting as crop pollinators and pest controllers.

Sadly, insect biodiversity is declining worldwide, with agriculture being a major contributor to this decline. For example, monoculture farming is a common form of farming where only one crop species is grown at a time, providing limited habitats and food resources for insects. By cultivating a diversity of crops in narrow, alternating strips, a technique known as strip cropping, farmers might make fields more suitable for insects, without reducing crop yield and productivity. However, so far, it was unknown if strip cropping can indeed increase insect biodiversity.

Croijmans et al. set out to investigate whether strip cropping can increase the biodiversity of ground beetles. Ground beetles play a key role in agricultural ecosystems, preying on common insect pests and weeds. They are sensitive to changes in farming practices and are often used as an indicator of agricultural sustainability. For this purpose, the researchers analysed four years of data from four organically managed experimental farms in the Netherlands, which included a diverse set of crops.

Croijmans et al. found that strip-cropped fields have more beetle species and more individual beetles than monocultures. As different ground beetle communities have natural preferences for specific crops, it is thought that the higher number of ground beetle species in strip-cropped fields is mostly due to the combination of two crop-related communities, rather than species unique to strip cropping. For example, if cabbage is strip-cropped with wheat, one would mostly find the ground beetle species corresponding to both cabbage and wheat, but few additional species. Interestingly, some ground beetle species preferred strip-cropped fields, while others preferred monocultures.

In conclusion, strip cropping can be a strategy to increase ground beetle biodiversity without losses to crop production. Therefore, farmers wanting to increase biodiversity might consider this approach instead of standard monocultures. Many biodiversity-increasing measures, such as flower strips or hedgerows, take up land that might otherwise be used for crop production. Strip cropping allows a more biodiverse field while keeping all land in production, making it a biodiversity measure that enables farmers to maintain the same level of crop production.

*Pilotto et al., 2020*; *Seibold et al., 2019*; *van Klink et al., 2024*) and in total species richness and changes in assembly composition (*Blowes et al., 2022*; *Wagner et al., 2021*). Insects are essential for crop production through their role in decomposition, pollination, pest control, and sustaining food webs. Therefore, erosion of insect communities can have potentially devastating effects on ecosystem functioning, provision of ecosystem services, and ultimately on human civilization (*Dirzo et al., 2014*). The main drivers of insect biodiversity decline are habitat loss due to conversion to agriculture, pollution, invasive species, and climate change (*Díaz et al., 2019*; *Müller et al., 2024*; *Wagner et al., 2021*). Strategies for biodiversity conservation in conjunction with adequate food production require understanding of how biodiversity responds to agricultural management (*Cozim-Melges et al., 2024*; *Mupepele et al., 2021*; *Saunders, 2020*). Ideally, sustainable agricultural practices retain yield and enhance biodiversity, preventing the need to convert natural habitats to agricultural land to maintain food production (*Tscharntke et al., 2021*).

Increasing crop heterogeneity can facilitate biodiversity conservation in highly productive agricultural landscapes without compromising yield (*Martin Guay et al., 2018*; *Sirami et al., 2019*). Crop diversification can enhance niche complementarity by creating heterogeneous habitats and increasing availability and diversity of resources (*Lichtenberg et al., 2017*; *Tamburini et al., 2020*). A promising crop diversification strategy is strip cropping, where crops are grown in alternating strips, wide enough for using standard agricultural machines yet narrow enough to facilitate ecological interactions among crops (*Ditzler et al., 2021*). Crops that are grown in strips may benefit from increased resource use efficiency and the suppression of pests and diseases (*Croijmans et al., 2024*; *Karssemeijer et al., 2024*; *Rakotomalala et al., 2023*) without major yield compromises (*Campanelli et al., 2023*; *Juventia and van Apeldoorn, 2024*; *van Oort et al., 2020*). Growing multiple crops on a field may foster a larger diversity of organisms than monocultures through greater plant species richness

that cascades into richer herbivore and predator communities (*Crutsinger et al., 2006*; *Cuperus et al., 2023*). Moreover, the expected increase in available and potentially complementary niches within the agricultural field due to higher spatial diversity in crops can result in admixture of communities related to individual crops (*Hummel et al., 2012*), or the creation of completely new communities by enhanced richness of agriculture-related species, and/or the occurrence of species rarely found in agricultural fields (*Rischen et al., 2021*). So far, it is not well understood if and how insect communities respond to strip cropping across distinct crops.

Here, we present data of 4 years of pitfall trapping of ground beetles at several moments during the growing season in 14 crops at four organic experimental farms across the Netherlands where strip cropping was compared to monocultures. Ground beetle communities are sensitive to changes in farming practices and are frequently used to examine agricultural sustainability (*Holland and Luff, 2000*; *Makwela et al., 2023*; *Turin, 2022*). Furthermore, ground beetle species are important for maintaining ecological functions as they comprise scavengers and predators of (weed) seeds, detritivores (e.g., collembolas and earthworms), and herbivores (e.g., aphids and caterpillars). We first examine whether strip cropping fields have greater ground beetle activity density, species richness, evenness, and diversity than monocultural fields. We also test for 12 abundant ground beetle genera whether their activity density is higher in strip cropping than in monocultures. Lastly, we evaluate whether ground beetle community changes are caused by admixture of communities, whether these assemblages promote species associated with agricultural or with natural ecosystems, and whether they contain rare species.

## Results

A total of 48,108 ground beetles belonging to 71 species were caught using pitfall traps over 4 years at four different organically managed experimental farms in The Netherlands: 40,153 at Almere; 3777 at Lelystad; 1126 at Valthermond; and 3052 at Wageningen (*Supplementary file 1 and 2*).

### Strip cropping enhances ground beetle richness

Strip cropping fields had on average 15% higher ground beetle taxonomic richness than monoculture fields after rarefaction to the number of samples of the least-sampled crop configuration ($\beta_0$=0.151, SE = 0.044, p<0.001; *Figure 1c and d*). However, strip cropping fields did not harbor more species than monocultural fields with the highest ground beetle richness ($\beta_0$=–0.008, SE = 0.037, p=0.821; *Figure 1d*). The difference in field-level taxonomic richness could not be explained by an increase in the number of ground beetle species per crop in strip cropping compared to monocultures. At crop level, the 5% increase in ground beetle taxonomic richness in strip cropping was not statistically significant (*Figure 1f*). Similarly, crop-level absolute evenness, inverse Simpson index, and Shannon entropy did not differ significantly among crop configurations (*Figure 1h–j*, *Figure 1—figure supplements 1 and 2*). The effect of crop configuration on crop-level taxonomic richness was variable and was not associated with location or crop species. The effect ranged from 56% more species in potato in monoculture at Wageningen in 2022 to 136% more species in barley in strip cropping at Wageningen in 2020 (*Figure 1k*).

### Strip cropping enhances ground beetle activity density

Ground beetle activity density was on average 30% higher in strip cropping fields than in monoculture fields ($\beta_0$=0.303, SE = 0.121, p=0.012; *Figure 1e*), based on rarefaction. However, there was no significant difference in activity density between the strip cropping fields and monocultures that harbored the richest beetle communities ($\beta_0$=–0.110, SE = 0.088, p=0.215). Crop-level activity density of ground beetles was not affected by crop configuration (*Figure 1g*, *Figure 1—figure supplement 1*).

### Crop configuration alters abundance of abundant genera

We tested how crop configuration affected the abundance of twelve abundant ground beetle genera (*Figure 2—source data 1*). We analyzed this separately per location as some genera only occurred at specific locations. Four genera were more abundant in strip-cropped fields in at least one location (*Anchomenus, Bembidion, Harpalus,* and *Nebria*), whereas four genera were more abundant in monocultures in at least one location (*Amara, Calathus, Pterostichus,* and *Trechus*) (*Figure 2*). The other

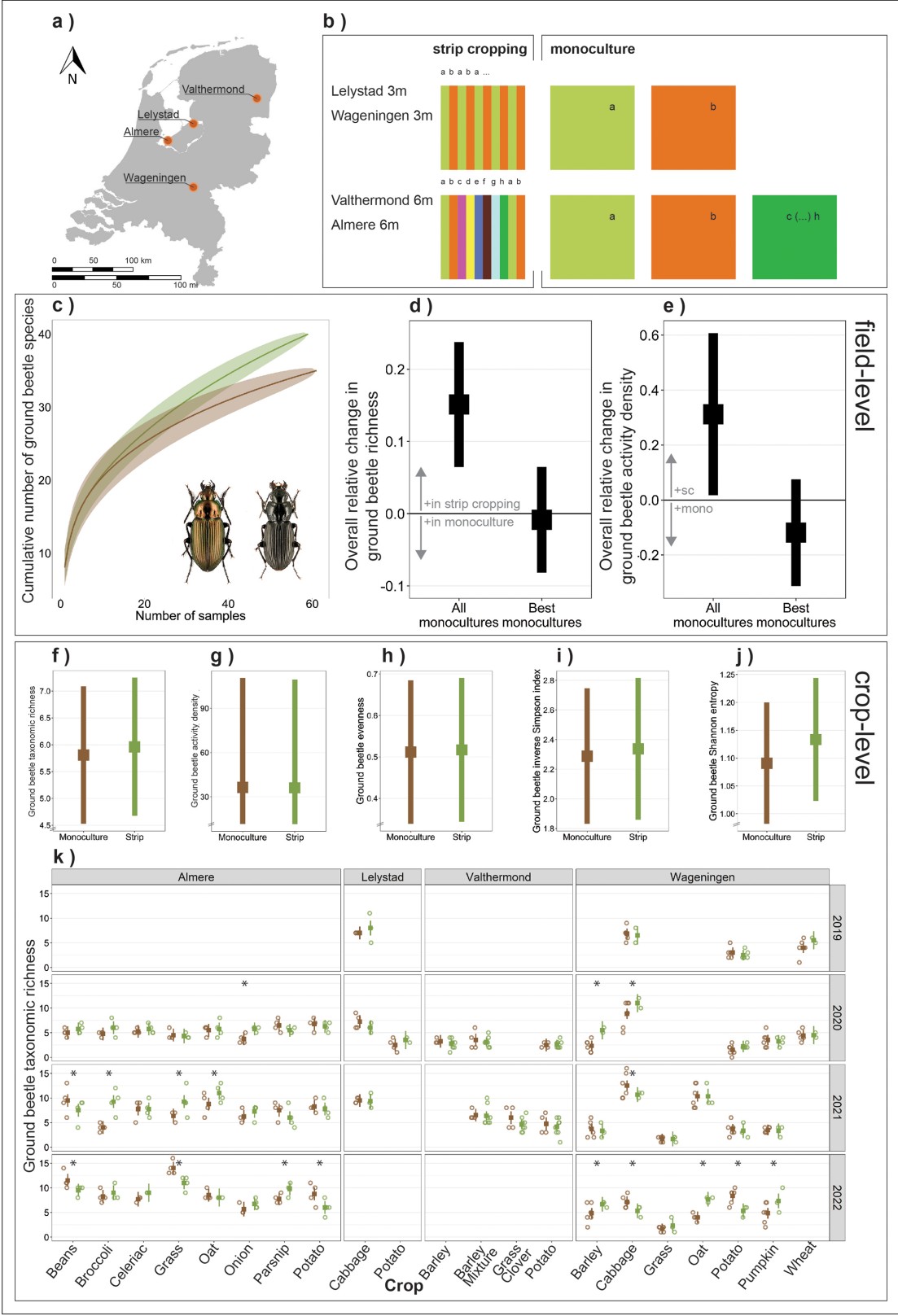

**Figure 1.** Effect of crop configuration (monoculture versus strip cropping) on ground beetle biodiversity. (**a**) Location of experimental sites in the Netherlands. (**b**) Field set-up of the two crop configurations: monoculture and strip cropping. At Lelystad and Wageningen, strip cropping consisted of 3-m-wide crop strips of two crops (pairs), and multiple crop pairs were assessed. At Almere and Valthermond, strip cropping consisted of 6-m-wide crop strips of eight crops combined (*Figure 1—figure supplements 3–14*). (**c**) Sample-based species accumulation curves of all year series from

*Figure 1 continued on next page*

*Figure 1 continued*

monocultures (brown) and strip cropping (green), in Almere from 2021 and 2022. This was the only location where an equal number of samples were taken in the monocultures and in strip cropping on a similar area. Ground beetle species include *Poecilus cupreus* (left) and *Pterostichus melanarius* (right). (**d, e**) Overall relative change in field-level ground beetle (**d**) taxonomic richness and (**e**) activity density. Positive values indicate higher richness or activity density in strip cropping, negative values in monocultures. (**f–j**) Overall relative effect of crop configuration on ground beetle (**f**) taxonomic richness, (**g**) activity density, (**h**) absolute evenness, (**i**) inverse Simpson index, and (**j**) Shannon entropy. (**k**) Effect of crop configuration on ground beetle taxonomic richness for each combination of location, year, and crop. Barley mixture consists of a mixture of barley-bean (2020) or barley-pea (2021). Squares indicate estimated means, the bar indicates the 95% confidence interval, both based on generalized linear mixed models. Asterisks indicate significant differences among the crop configurations. Empty panels indicate combinations of years and locations that were not sampled. When no estimated mean and confidence interval are shown, crops were not grown or sampled in that year. Open circles indicate individual year series to visualize sample size (***Supplementary file 6***).

© 2011, Ortwin Bleich. Figure 1 panel c ground beetle images were reprinted from http://www.eurocarabidae.de/ with permission from Ortwin Bleich. It is not covered by the CC-BY 4.0 license and further reproduction of this panel would need permission from the copyright holder.

The online version of this article includes the following figure supplement(s) for figure 1:

**Figure supplement 1.** Effect of crop configuration on ground beetle activity density and absolute evenness.

**Figure supplement 2.** Effect of crop configuration on ground beetle inverse Simpson index and Shannon entropy.

**Figure supplement 3.** Field map of experimental lay-out at Almere in 2020.

**Figure supplement 4.** Field map of experimental lay-out at Almere in 2021.

**Figure supplement 5.** Field map of experimental lay-out at Almere in 2022.

**Figure supplement 6.** Field map of experimental lay-out at Lelystad in 2019.

**Figure supplement 7.** Field map of experimental lay-out at Lelystad in 2020.

**Figure supplement 8.** Field map of experimental lay-out at Lelystad in 2021.

**Figure supplement 9.** Field map of experimental lay-out at Valthermond in 2020.

**Figure supplement 10.** Field map of experimental lay-out at Valthermond in 2021.

**Figure supplement 11.** Field map of experimental lay-out at Wageningen in 2019.

**Figure supplement 12.** Field map of experimental lay-out at Wageningen in 2020.

**Figure supplement 13.** Field map of experimental lay-out at Wageningen in 2021.

**Figure supplement 14.** Field map of experimental lay-out at Wageningen in 2022.

four common genera (*Blemus, Clivina, Loricera,* and *Poecilus*) were not significantly influenced by crop configuration. Furthermore, no ground beetle genus showed significantly contrasting responses to crop configuration among locations.

## Crop configuration alters ground beetle community composition

Ground beetle communities were significantly influenced by crop configuration, but the effects were highly dependent on the specific context created by the combination of location, year, and crop (***Table 1***, Appendix 1, ***Supplementary file 3***). The context dependency of configuration effects on ground beetle communities is illustrated in redundancy analyses of ground beetle assemblages per crop combination at Wageningen in 2021 and 2022 (Appendix 2). Here, we found distinct ground beetle communities among crop configurations for pumpkin, barley, and potato in 2021, and for cabbage and oat in 2022. In the other cases, the difference between crop configurations was not significant (Appendix 2, ***Supplementary file 4***). Moreover, in all crop combinations except for potato-grass in 2021, the difference in ground beetle communities between the monocultures of the constituent crops was significant, while this was never the case for ground beetle communities of crops in strip cropping (***Supplementary file 4***). This indicates that strip cropping might lead to overlapping crop-related communities. However, these results could be spatially autocorrelated as samples from different crops were in closer proximity of each other in strip cropping than among monocultures (***Figure 1—figure supplements 13 and 14***).

## Crop configuration does not increase the number of rare ground beetle species

Among the 461 year series, we only found two rare species (following waarneming.nl): one individual of *Microlestes minutulus* and five individuals of *Harpalus signaticornis*. This latter species was

found most often in a wheat monoculture, but this was likely due to a failed crop which created a very open habitat. *H. signaticornis* is known to inhabit recently disturbed, dry, open habitats with limited vegetation (*Turin et al., 2012*), a situation similar to this sparsely covered monoculture. All other species were common or relatively common species (*Supplementary file 2*). Furthermore, most ground beetle species were either ruderal habitat specialists or eurytopic species that occur in many different habitats (*Supplementary file 2*).

## Discussion

It is well established that different crop types have distinct ground beetle communities (*Eyre et al., 2009*; *Holland and Luff, 2000*) and that increasing habitat diversity by including multiple crops in a field can enhance ground beetle diversity (*Cuperus et al., 2024*; *Puliga et al., 2023*). Our study shows that strip cropping increased field-level ground beetle richness by 15%. However, ground beetle communities within the same crop in a strip or in monoculture were mostly similar. This indicates that the 15% increase in richness at the field level can be mostly attributed to the higher number of crops in strip-cropped fields that harbored crop-related ground beetle communities, and that there was only limited mixing of ground beetle species among crops in strip crops. This is in line with earlier findings that ground beetle movement is reduced by crop edges (*Allema et al., 2014*; *Anderson et al., 2024*). Further research on movement behaviors of ground beetles at crop edges might help explain how ground beetles distribute themselves within a strip cropping field, and whether they utilize the different resources provided by a more diverse cropping system.

The 30% higher ground beetle activity density in strip cropping fields compared to monoculture fields may be explained by a more stable and diverse habitat with refuges and alternative resources in strip crops (*Ratnadass et al., 2012*). Crop diversification enhances prey biomass for ground beetles (*Lichtenberg et al., 2017*) and increases weed seed richness as compared to monocultures (*Ditzler et al., 2023*), both of which reduce bottom-up control by increased food provision (*Carbonne et al., 2022*). Alternatively, the increase in activity density might be caused by higher movement of ground beetles, which in turn can be the consequence of food starvation (*Wallin and Ekbom, 1994*). Indeed, several papers show reduced abundances of herbivorous insects in strip- and intercropping (*Alarcón-Segura et al., 2022*; *Cuperus et al., 2023*; *Rakotomalala et al., 2023*), which may be potential prey for ground beetles (*Turin, 2000*). Furthermore, strip cropping systems may support high herbivore reproduction in combination with a high predation pressure resulting in herbivore populations dominated by early life stages (*Karssemeijer et al., 2024*). Therefore, strip cropping may favor ground beetles that predate on, for instance, herbivore eggs, such as *Bembidion* spp. and *Anchomenus dorsalis*, which were more abundant in strip cropping (*Finch, 1996*), and disadvantaging ground beetles that feed on larger prey, such as *Pterostichus melanarius,* which was less abundant in strip

**Table 1.** Effect of crop configuration on ground beetle community composition.
Results from permanova analysis using Hellinger's transformation for data from the three locations with species-level data (see *Supplementary file 3* for analyses per location). Crops were a nested variable within years, as these differed among years. Years were nested in locations as the years that were studied differed among locations. Bold indicates significant effects ($\alpha$=0.05).

| Predictor | Df | Sum sq | R² | F | p |
|---|---|---|---|---|---|
| Crop configuration | 1 | 1.68 | 0.01 | 7.32 | **0.001** |
| Location | 2 | 47.7 | 0.29 | 103.8 | **0.001** |
| Location: Year | 6 | 10.7 | 0.06 | 7.72 | **0.001** |
| Location: Year: Crop species | 31 | 33.3 | 0.20 | 4.67 | **0.001** |
| Crop configuration: Location | 2 | 0.63 | 0.00 | 1.37 | 0.149 |
| Crop configuration: Location: Year | 6 | 2.31 | 0.01 | 1.67 | **0.012** |
| Crop configuration: Location: Year: Crop species | 31 | 14.8 | 0.09 | 2.08 | **0.001** |
| Residual | 245 | 56.3 | 0.34 | | |
| Total | 324 | 167.4 | 1.00 | | |

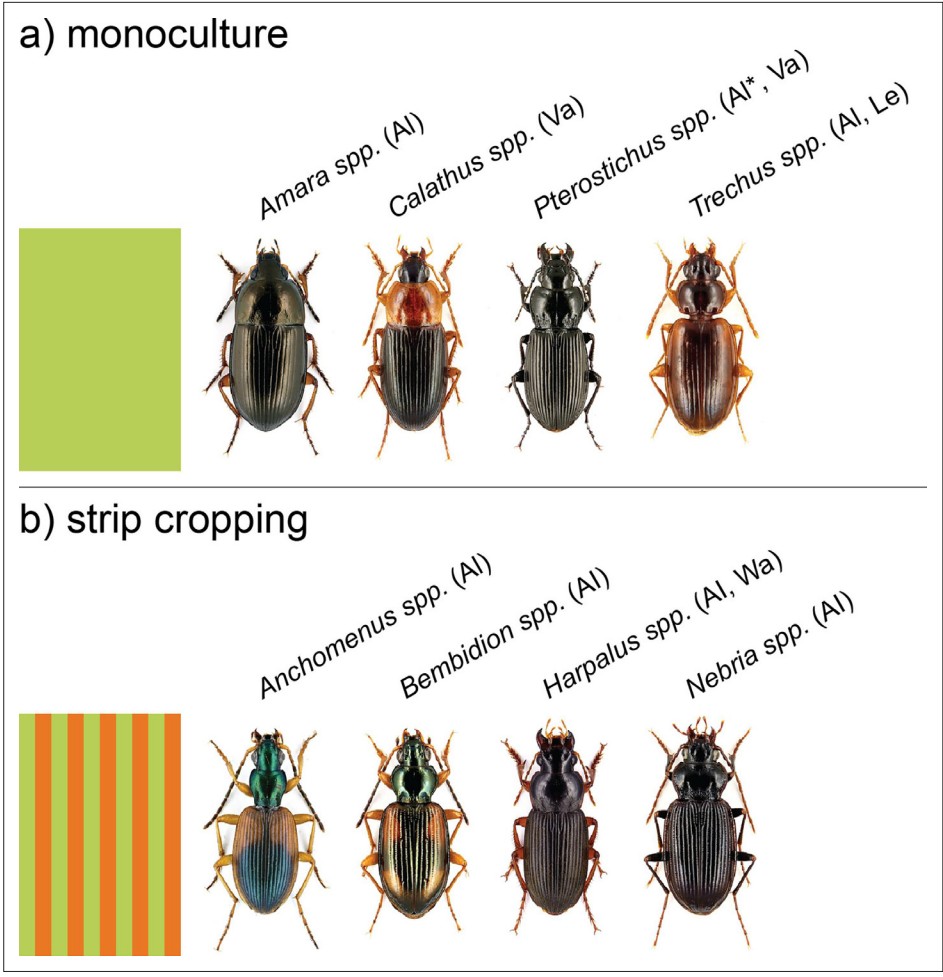

**Figure 2.** Ground beetle species associated with crop configuration (monoculture [**a**] versus strip cropping [**b**]). Results obtained by generalized linear mixed models on the twelve most common genera of ground beetles with the four locations analyzed separately (*Figure 2—source data 1*). Only those genera are given for which cropping system significantly influences activity density within at least one location (α=0.05). Locations where the genus had a higher activity density in one of the cropping systems are indicated between brackets (Al = Almere; Le = Lelystad; Va = Valthermond; Wa = Wageningen). For Pterostichus, data from Almere in 2020 was analyzed separately, as models did not fit elsewhere. The asterisk '*' next to Pterostichus indicates that only in 2020 the difference between monoculture and strip cropping was significant.

The online version of this article includes the following source data for figure 2:

**Source data 1.** Abundances of the 12 most abundant ground beetle genera in monoculture and strip cropping fields in four locations.

cropping. Multi-taxa evaluation of future strip cropping studies will be a valuable approach to increase understanding of biomass flows and trophic interactions within diverse agricultural systems.

The 15% increase in ground beetle species richness through strip cropping is mostly caused by an increase in species (relatively) common to agricultural fields, rather than an increase in rare species or species otherwise common in other habitat types. A keystone species here might be *P. melanarius*, the most common *Pterostichus* species in our samples. *P. melanarius* can be very dominant in pitfall traps, as we regularly found hundreds of individuals per trap. This relatively large ground beetle might compete with or predate on other ground beetle species (*Roubinet et al., 2018*). *P. melanarius* is especially well-adapted to highly intensive agriculture (*Turin, 2022*; *Turin, 2000*), which explains

why it was more abundant in monocultures previously (*Ditzler et al., 2021*). In turn, the reduced *P. melanarius* populations in strip cropping might allow other species to persist, such as *Harpalus* spp. and *A. dorsalis*. Therefore, strip cropping might facilitate ground beetle biodiversity by reducing the dominance of a common, competitive species.

Biodiversity gains in our study did in most cases not coincide with productivity losses (*Supplementary file 5*). Earlier studies on crop yield in the strip cropping fields in Lelystad and Wageningen during the years of pitfall trapping show a yield decrease when strip cropping cabbage (*Carrillo-Reche et al., 2023*) and wheat (*Ditzler et al., 2023*), whereas potato yield was unaffected by crop configuration (*Ditzler et al., 2023*). In Almere, bean and parsnip yields were higher in strip cropping, whereas oat and onion yields were lower (*Juventia and van Apeldoorn, 2024*). In Valthermond, crop productivity was similar for monocultures and strip cropping (*Supplementary file 5*). Therefore, biodiversity gains through strip cropping do not compromise agricultural production.

We show that changing crop configuration from monoculture to strip cropping, on average, enhances ground beetle richness by 15% and activity density by 30% within agricultural fields. These results show that strip cropping can lead to increases in biodiversity that approach those achieved by shifting from conventional to organic farming practices (+19% richness, *Lichtenberg et al., 2017*; +23% richness, *Gong et al., 2022*) and by other in-field diversification measures, like hedgerows and flower strips (+23% richness, *Lichtenberg et al., 2017*; +24% richness, *Beillouin et al., 2021*). While organic management or in-field diversification measures generally lead to lower productivity (*Gong et al., 2022*), strip cropping can maintain crop productivity without taking land out of production (*Campanelli et al., 2023*; *Juventia and van Apeldoorn, 2024*; *van Oort et al., 2020*), although occasional yield reductions have been reported (*Carrillo-Reche et al., 2023*; *Ditzler et al., 2023*). The biodiversity gain in our study was achieved without additional crop-level diversification strategies designed for biodiversity conservation, such as cover cropping or (flowering) companion plants. This increase in biodiversity stacks on top of already higher biodiversity achieved through organic management (*Gong et al., 2022*; *Lichtenberg et al., 2017*). Furthermore, the estimated richness effects of strip cropping, that is, no change at crop level and 15% increase at field level, are conservative because they only consider the pairwise comparison of biodiversity of two or three crops in a strip cropping configuration to monocultures, whereas the inclusion of more crops within strip cropping could further enhance ground beetle biodiversity. Also, the effect of strip cropping on ground beetle biodiversity might be more pronounced in conventional agriculture and at larger field sizes, as the relatively small-scale, organic fields that we studied might have already had a relatively high ground beetle biodiversity (*Tscharntke et al., 2021*). The ground beetle communities in our study were dominated by farmland and eurytopic species and contained only two rare species (*Turin, 2022*). Future research could test whether the inclusion of other in-field diversification measures within strip-cropped fields, such as the establishment of perennial semi-natural habitats (SNHs) (*Rischen et al., 2021*; *Sirami et al., 2019*) or uptake at larger spatial extents (*Tscharntke et al., 2021*) would allow ground beetles with other habitat preferences to establish in agricultural fields.

## Materials and methods
### Study area
A multi-location study was conducted on four organic farms across the Netherlands (*Figure 1a*). Three experimental farms were managed by Wageningen University & Research (Lelystad, Valthermond, Wageningen) and one commercial farm was managed by *Exploitatie Reservegronden Flevoland* located in Almere. All four locations contained both strip cropping and monocultural crop fields, but differed in soil type, establishment year of the strip cropping experiment, number of crops grown, length of the crop rotation, number of sampled crops and sampling years, and farm and landscape characteristics such as percentage of on-farm SNH, mean field size, and landscape configuration (*Figure 1b*, *Figure 1—figure supplements 3–14*). The locations Almere and Lelystad were located in a homogeneous, open polder landscape characterized by intensive arable crop production and non-crop habitats consisting of grass margins, tree lines, and watercourses. Valthermond was located in an open, reclaimed peat landscape with intensive arable crop production characterized by long and narrow fields separated by grassy margins and ditches and limited areas of woody elements. The site

at Wageningen was located in a more complex landscape with smaller field sizes and non-crop habitat consisting of woodlots, hedgerows, tree lines, ditches, and farmyards.

## Experimental layout

At Almere and Valthermond, the crops in strip cropping were all grown alongside each other, whereas at Lelystad and Wageningen two alternating crops (crop pairs) were grown alongside each other (*Figure 1b*, *Figure 1—figure supplements 3–14*). At each location, strip cropping and monoculture fields were always paired on the same experimental field. At Almere, eight different crops were grown in alternating strips of 6 m width, including celeriac (*Apium graveolens* var. *rapaceum*), broccoli (*Brassica oleracea* var. *italic*), oat (*Avena sativa*), onion (*Allium cepa*), parsnip (*Pastinaca sativa*), faba bean (*Vicia faba*), potato (*Solanum tuberosum*), and a mix of ryegrass and white clover referred to as grass-clover (*Lolium perenne/Trifolium repens*) (*Figure 1—figure supplements 3–5*). At Lelystad, four different crop pairs were grown in alternating strips of 3 m width, including carrot (*Daucus carota* subsp. *sativus*) and onion, white cabbage (*Brassica oleracea* var. *capitata*) and wheat (*Triticum aestivum*), sugar beet (*Beta vulgaris*) and barley (*Hordeum vulgare*), and potato and ryegrass (*Figure 1—figure supplements 6–8*). At Valthermond, eight different crops were grown in alternating strips of 6 m width, including potato, barley, barley mixed with broad bean (*V. faba*) in 2020, barley mixed with pea (*Pisum sativum*) in 2021, sweet corn (*Zea mays convar. saccharata* var. *rugosa*), sugar beet, common bean (*Phaseolus vulgaris*), and grass-clover (*Figure 1—figure supplements 9 and 10*). At Wageningen, three different crop pairs were grown in alternating strips of 3 m width, including white cabbage and wheat (2019–2021) or oat (2022), barley and pumpkin (*Cucurbita maxima*, 2020–2022) or bare soil due to crop failure (2019), and potato and ryegrass (*Figure 1—figure supplements 11–14*). The crop combinations and neighbors were selected based on literature, expert knowledge, and experience of functionality in terms of expected advantages for yield and pest and disease control. Large-scale monoculture plots (0.25–2.30 ha) served as reference, hereafter referred to as monoculture. At Lelystad, Valthermond, and Wageningen, not each crop grown in strips was present as monoculture in each year, but only those crops for which a monoculture was present were sampled (*Figure 1k*, *Supplementary file 6*). All fields were managed according to organic regulations, yet at each location fertilization and weed management reflected regional practices and were adjusted to local soil conditions. Flower strips were sown within the experimental fields at Almere and Valthermond (*Supplementary file 7*).

## Sampling

The ground beetle community was sampled using pitfall traps in all crops for which both a monoculture and strip cropping field were present, at each location and in multiple rounds per year between March and September. The sampled crops, number of rounds, and number of pitfalls differed per year and per location (*Supplementary file 6*). Pitfall traps consisted of a transparent plastic cup (9.2 cm diameter, 14 cm height) placed in the soil so the top of the cup was flush with the soil surface. We did not use funnels inside the cups. Pitfalls were filled with approximately 100 ml water mixed with non-perfumed soap (around 3 cm high water level) and covered with a black plastic rain guard (12.5 cm diameter) around 2–5 cm above the soil surface. Pitfall traps were placed in the center of a strip (1.5–3 m from the edges of the strip), and at least 10 m from field edges at fixed locations at different moments in a year. Furthermore, traps were mostly placed at equal distances from the field edges in strip-cropped and monocultural fields. At Almere and Valthermond, the number of pitfalls in monocultures and strip cropping was the same. At Wageningen and Lelystad, the number of pitfalls in strip-cropped fields was usually lower than in the monoculture because the area of the strips was only half of that of the monocultures.

## Ground beetle identification

Ground beetles were identified up to species level for Almere in 2021 and 2022, Lelystad and Wageningen. At Almere in 2020, identification was kept at the genus level due to an extremely high abundance of *P. melanarius/niger* and *Poecilus cupreus/versicolor*, and the associated high time investment for identification. At Valthermond, due to labor constraints, we chose to consider species that are complicated to distinguish only up to genus level (like *P. cupreus/versicolor*, *Harpalus* spp. other than *H. rufipes* and *H. affinis*, *Bembidion* spp.). For all analyses, year series were made, in which all ground

beetle catches from the same pitfall trap were pooled per year. We examined habitat preferences of ground beetle using *Turin, 2022* and ground beetle rarity using https://waarneming.nl/ (checked on January 15, 2025).

## Statistical analyses

We used R, version 4.2.2, for all statistical analyses.

### Effect of crop configuration on field-level richness and activity density

To analyze the difference in species richness and activity density between monocultures and strip cropping configurations at field level, we used rarefaction of samples within the same field. To rarify to an equal sampling intensity, we calculated the average cumulative number of species or individuals within x year series, where x is the largest number of year series available for the crop configuration comparison (correcting for unequal sampling between monocultures and strip cropping, or for missing samples). Next, we calculated the relative change due to strip cropping by subtracting the number of species or individuals found in the monoculture field from the number found in the strip cropping field and then dividing the result by the number of species or individuals in the monoculture (*Zou et al., 2020*). This gave the relative change centered around zero, where negative values indicated higher richness in monocultures and positive values higher richness in strip cropping. We then analyzed this data using generalized linear mixed models (GLMM) with a Gaussian distribution and assessed whether the intercept deviated significantly from zero. As random variables, we used location and year, with year nested in location. We ran these analyses using a dataset that included all comparisons among monocultural fields and strip cropping fields of all locations. However, as in this case the strip-cropped field was compared with both monocultural fields of the corresponding crop pair, we also tested the effect of field-level richness using only one monoculture per strip-cropped field. To obtain a conservative estimate of the effect of strip cropping, we chose the monocultural fields with the highest taxonomic richness or activity density among the constitutive crops of the strip cropping fields. Generalized linear models (GLMs) were run using the glmmTMB package (*Brooks et al., 2017*) and tested for model fit using the DHARMa package (*Hartig, 2018*).

### Effect of crop configuration on crop-level biodiversity

To quantify biodiversity we used five variables: (1) activity density, the total number of ground beetles found per year series; (2) taxonomic richness, the total number of species or genera (lowest taxonomic level available) found per year series; (3) the inverse Simpson index, the inverse of the sum of proportions of different species over the total abundance (*Simpson, 1949*); (4) absolute evenness, the number of effective species calculated by dividing the inverse Simpson index by the taxonomic richness (*Williams, 1964*); and (5) Shannon entropy (*Shannon and Weaver, 1949*). We chose absolute evenness as our measure for evenness as this method removes the richness component from the inverse Simpson index and adheres to all requirements for an evenness index (*Smith and Wilson, 1996*; *Tuomisto, 2012*). We included both the inverse Simpson index and Shannon entropy as the former is more sensitive to changes in evenness and the latter to species richness (*DeJong, 1975*).

To analyze the effect of crop configuration on total ground beetle activity density, taxonomic richness, evenness, inverse Simpson index, and Shannon entropy, we used GLMM. We constructed models for each response variable, using data from all four locations. In these models, we included crop configuration (monoculture or strip cropping) as a fixed factor. We included location, year, and crop as nested random variables in these models, with crop nested in year and year in location. To quantify and visualize the variation in responses between locations, years, and crops, we ran GLMs with a variable that combined these three variables into one, which was also included as a fixed factor. Here, we also included the interaction between crop configuration and the combined variable for crop, location, and year. For the model on activity density, we used the negative binomial distribution (log link function), as this was count data; for richness, evenness, and Shannon entropy, we used Gaussian distribution; and for the inverse Simpson index, we used a gamma distribution (inverse link function) as a Gaussian distribution did not fit well. We used the 'DHARMa' package to validate model assumptions (*Hartig, 2018*). We used estimated marginal means to assess differences between monoculture and strip cropping (*Hothorn et al., 2016*; *Lenth, 2018*).

To analyze the effect of crop configuration on the activity density of the 12 most common ground beetle genera, we used GLMMs with negative binomial distribution. Fixed effects were crop configuration, location, and their interactions, and crop nested in year was added as a random effect. As certain genera only occurred at specific locations, we only included locations where the genus was found. All models were tested for model fit using the DHARMa package. We used estimated marginal means to assess differences between monoculture and strip cropping.

## Community composition of crop configurations and crops

To assess whether crop configurations have distinct ground beetle communities, we used permanova with a Hellinger transformation (with 999 permutations) using the 'vegan' package (*Oksanen et al., 2013*). We used a Hellinger transformation to give more weight to rarer species, thus accounting for highly abundant species in some samples (such as *P. melanarius* and *H. rufipes*). We only included species that occurred in at least 3% of the pitfall samples to avoid strong influence of very rare species. We used data from locations and years where pitfall catches had been identified to species level (Almere 2021/22, Lelystad and Wageningen). We considered four models, one for all locations combined and one for each location separately. In all models, we included crop configuration and the nested variables of location (whenever applicable), year, and crop as fixed factors. We also analyzed the interaction between crop configuration and this nested structure of location, year, and crop. To visualize these results, we used non-metric multidimensional scaling (NMDS) on Hellinger transformed data. To show whether crop configurations have distinct ground beetle communities, we used redundancy analysis (RDA) on Hellinger transformed data. Here, we again conducted four analyses, one for all locations combined and one for each of the three considered locations. We only used crop configuration as a predictor to force RDA to show any change in ground beetle community associated with crop configuration. As such, only one RDA axis was created per model, which was plotted against the first principal component describing the residual variation.

Due to the large influence of location and year on ground beetle communities, visualizing any general effects of strip cropping on these communities using all available data was challenging. To address this, we conducted RDA and visualized the effect of crop configuration on a subset of the data from one location and one year. We chose the data from Wageningen in 2021 and 2022 for this analysis because it provided a set-up where strip cropping of two crops could be compared with their constituent monocultures within the same experimental fields. Here, we used both crop configuration, crop, and their interaction term as explanatory variables for each crop pair separately. To analyze whether ground beetle communities significantly differed among combinations of crops and crop configurations, we ran pairwise permanova on all three fields and two years separately using the 'pairwiseAdonis' package (*Martinez, 2020*).

## Acknowledgements

First, we are grateful to Ron Anbergen, Martine Arkema, Bart Burger, Jonas Driessen, Michiel van de Glind, Angelo Grievink, Roelof Gruppen, Rolinde de Haan, Willem Hendriks, Nashita Maniran, Marina Martino, Sara Michellin, Ciska Nienhuis, Ralph Rustom, Simone Verdonschot, Pim Vrehen, Rik Waenink, and Xiaoshen Wang for their contribution to data collection. This work could not have been completed without the occasional help of many colleagues and students. For this we thank Zhaoqi Bin, Lenora Ditzler, Hilde Faber, Merel Hofmeijer, Gabriel Joachim, Stella Juventia, Peter Karssemeijer, Nelson Ríos Hernández, and any other people that helped. Thanks go to Roy Michielsen and Dirk van de Weert for maintaining the Almere fields; the team at Wageningen Field crops, and in particular Joost Rijk and Laurens van Run for maintaining the Lelystad fields; Gerard Hoekzema for maintaining the Valthermond fields; Olivia Elsenpeter, Esther Hofkamp, Titouan le Noc, and the Unifarm staff, and in particular Andries Siepel and Peter van der Zee, for maintaining the Wageningen fields. We thank Ortwin Bleich for allowing us to use his ground beetle pictures. We are grateful to Daan Mertens, Marcel Dicke, Liesje Mommer, and Thijs Fijen for constructive feedback on earlier versions of this manuscript. Lastly, we thank editors Bernhard Schmid and Sergio Rasmann, and three anonymous reviewers for their extensive and constructive feedback. This study has received funding from the European Union's Horizon 2020 research and innovation program under grant agreements No 727482 (DiverIMPACTS) and No 727672 (LegValue), from the Dutch Public Private Partnership research program under grant agreement No LWV19129 (Crop diversification), from regional funds provided by provinces

Groningen and Drenthe, and through internal Wageningen University and Research funds financed by the Dutch Ministry of Agriculture, Nature and Food Quality under grant agreement No KB36003003 (Nature Based Solutions in Field Crops). This publication is part of the project CropMix (with project number NWA.1389.20.160) of the Dutch Research Agenda (NWA-ORC) which is (partly) financed by the Dutch Research Council (NWO).

## Additional information

### Funding

| Funder | Grant reference number | Author |
|---|---|---|
| European Commission | 727482 | Dirk F van Apeldoorn Walter AH Rossing |
| European Commission | 727672 | Dirk F van Apeldoorn |
| Dutch Public Private Partnership | LWV19129 | Dirk F van Apeldoorn |
| Ministry of Agriculture, Nature and Food Quality | KB36003003 | Fogelina Cuperus |
| Dutch Research Council | NWA.1389.20.160 | Fogelina Cuperus Erik H Poelman |

The funders had no role in study design, data collection and interpretation, or the decision to submit the work for publication.

### Author contributions
Luuk Croijmans, Fogelina Cuperus, Conceptualization, Data curation, Formal analysis, Supervision, Investigation, Visualization, Methodology, Writing – original draft, Writing – review and editing; Dirk F van Apeldoorn, Conceptualization, Supervision, Funding acquisition, Visualization, Project administration, Writing – review and editing; Felix JJA Bianchi, Conceptualization, Writing – original draft, Writing – review and editing; Walter AH Rossing, Conceptualization, Funding acquisition, Writing – original draft, Project administration, Writing – review and editing; Erik H Poelman, Conceptualization, Supervision, Funding acquisition, Methodology, Writing – original draft, Project administration, Writing – review and editing

### Author ORCIDs
Luuk Croijmans https://orcid.org/0000-0001-7690-8988
Fogelina Cuperus https://orcid.org/0000-0003-0616-7920
Dirk F van Apeldoorn https://orcid.org/0000-0003-0636-1977
Felix JJA Bianchi https://orcid.org/0000-0001-5947-9405
Walter AH Rossing https://orcid.org/0000-0003-2294-2368
Erik H Poelman https://orcid.org/0000-0003-3285-613X

Joint Public Review: https://doi.org/10.7554/eLife.104762.4.sa1
Author response https://doi.org/10.7554/eLife.104762.4.sa2

## Additional files

### Supplementary files
Supplementary file 1. All ground beetle species found among the four locations, and their species codes as used in several figure supplements.

Supplementary file 2. Total number of ground beetles caught per species (or genus), per location. The number of year series is given per location in brackets. For some locations, ground beetles were identified up to genus level, these are underlined. 'N/A' indicates that this taxa was identified to a different taxonomic level for the specific location. Locations are indicated with abbreviations (Al=Almere; Le = Lelystad; Va = Valthermond; Wa = Wageningen). Rarity indicates the rarity of the

species according to waarneming.nl (1 = common, 2 = relatively common, 3 = rare, 4 = very rare). Affinity indicates the habitat affinity group as by table A.1 of Turin et al. (2022), a * here indicates that species are more eurytopic.

Supplementary file 3. Effect of crop configuration on ground beetle community composition. Results from permanova analyses using Hellinger's transformation for data from the three locations with species-level data. 'Crop species' is a nested variable within years, as these differed among years. Years were nested in locations, as the years that were studied differed among locations. p-values in bold indicate significant effects (α = 0.05).

Supplementary file 4. Effect of crop configuration and crop species on ground beetle community composition. Results from pairwise permanova analyses for crop pairs pumpkin-barley, cabbage-oat, and potato-grass in 2021 (a) and 2022 (b) in Wageningen. Values show F-values for the comparison between the crop configurations and crops in crossing rows and columns. Bold numbers indicate significant differences between combinations of crop configurations and crops (α = 0.05).

Supplementary file 5. Effect of crop configuration on crop yield. Yield results were retrieved from published and unpublished studies on effects of strip cropping on crop yield in similar locations, years, and crops as this study. Mean crop yield is presented in ton per hectare (t/ha). When known, standard deviations of mean crop yield are given (± SD). When a crop is indicated with NC (not collected), the crop yield was not collected due to an inconsistent sampling method (potato, 2020, Almere), crop failure (broccoli, 2020, Almere; celeriac, 2021, Almere), unavailable machine-harvest data (grass, Almere, 2020, 2021, 2022), and undocumented reasons (barley/beans, 2020, Valthermond). Unavailable data include cabbage (2019) and potato (2020) in Lelystad; and pumpkin (2020, 2021, 2022), barley (2020, 2021, 2022), oat (2021, 2022), potato (2021, 2022), grass (2021, 2022), and cabbage (2022) in Wageningen.

Supplementary file 6. The total number of pitfall traps placed per location, year, and crop. Rounds indicate the number of times pitfall traps were placed and were pooled within year series.

Supplementary file 7. Plant species composition of flower strips adjacent to the strip cropping fields in Lelystad and Valthermond.

MDAR checklist

## Data availability
Data and scripts are publicly available from 4TU.ResearchData.

The following dataset was generated:

| Author(s) | Year | Dataset title | Dataset URL | Database and Identifier |
|---|---|---|---|---|
| Croijmans L, Cuperus F, van Apeldoorn DF, Bianchi FJJA, Rossing WAH, Poelman EH | 2025 | Data underlying the publication: Strip cropping shows promising increases in ground beetle community diversity compared to monocultures | https://doi.org/ 10.4121/bcf78320-aaa6-428f-acf6-2eb436baa13e | 4TU.Research Data, 10.4121/bcf78320-aaa6-428f-acf6-2eb436baa13e.v3 |

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

## Appendix 1

### The effect of crop configuration on ground beetle community composition using redundancy analysis (RDA) and non-metric multidimensional scaling (NMDS)

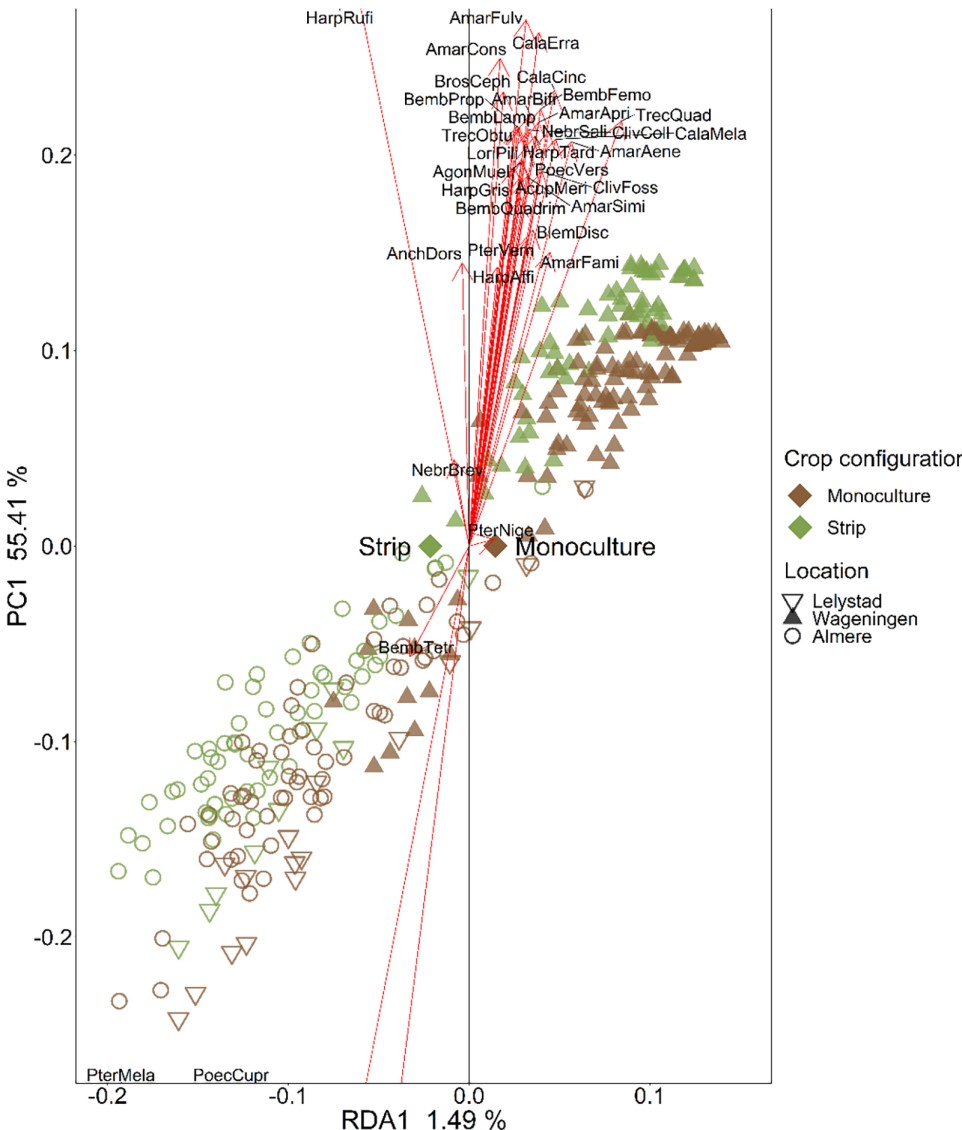

**Appendix 1—figure 1.** The effect of crop configuration on ground beetle community composition using redundancy analysis (RDA). Data from all datasets that included species-level data (Almere [O], Lelystad [∇], Wageningen [▲]) are used for visualizing the first RDA and PC axis. Each axis shows the percentage explained variation. Color of the dots indicates crop configuration (brown = monoculture, green = strip cropping), and the direction of each crop configuration on the RDA axis (x-axis) is mentioned. Red arrows indicate species placement and name codes are given close to the tip of the arrow, meaning of the name codes can be found in .

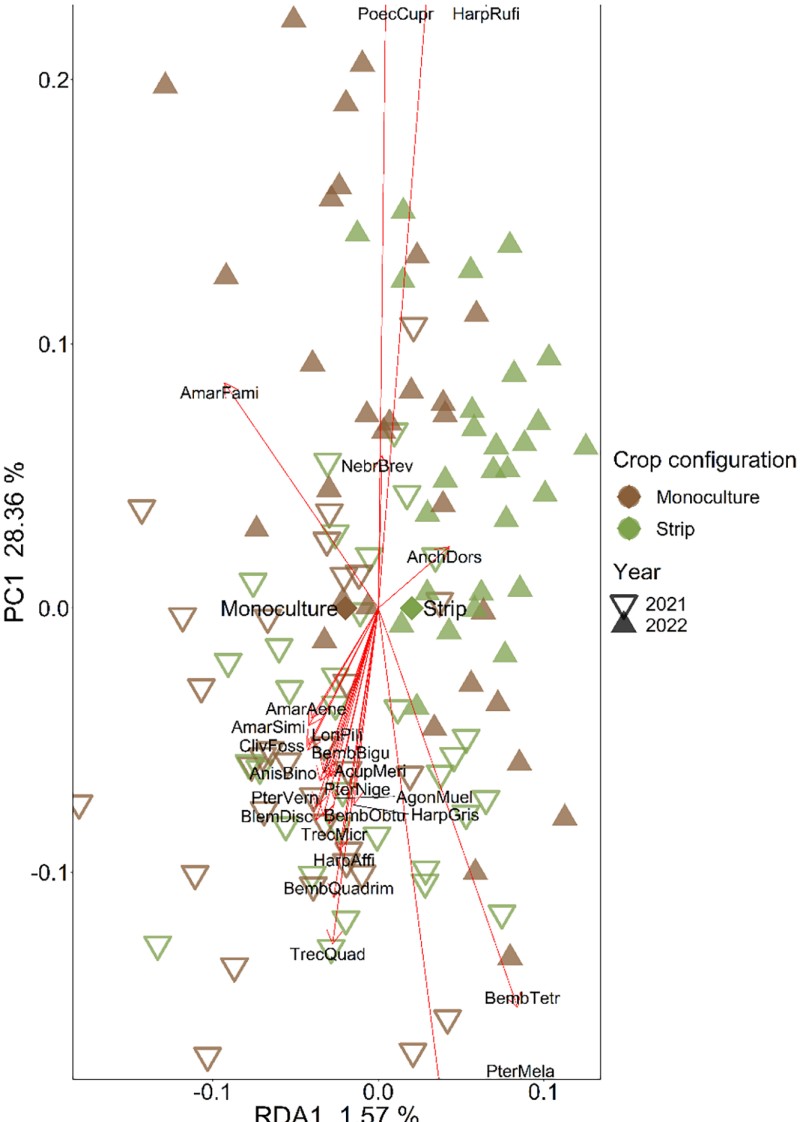

**Appendix 1—figure 2.** The effect of crop configuration on ground beetle community composition in Almere, using redundancy analysis (RDA). The first RDA and PC axis are visualized. Each axis shows the percentage explained variation. Shape of the dots indicates year (▽=2021, ▲=2022), color of the dots indicates crop configuration (brown = monoculture, green = strip cropping), and the direction of each crop configuration on the RDA axis (x-axis) is mentioned. Red arrows indicate species placement and name codes are given close to the tip of the arrow, meaning of the name codes can be found in *Supplementary file 1*.

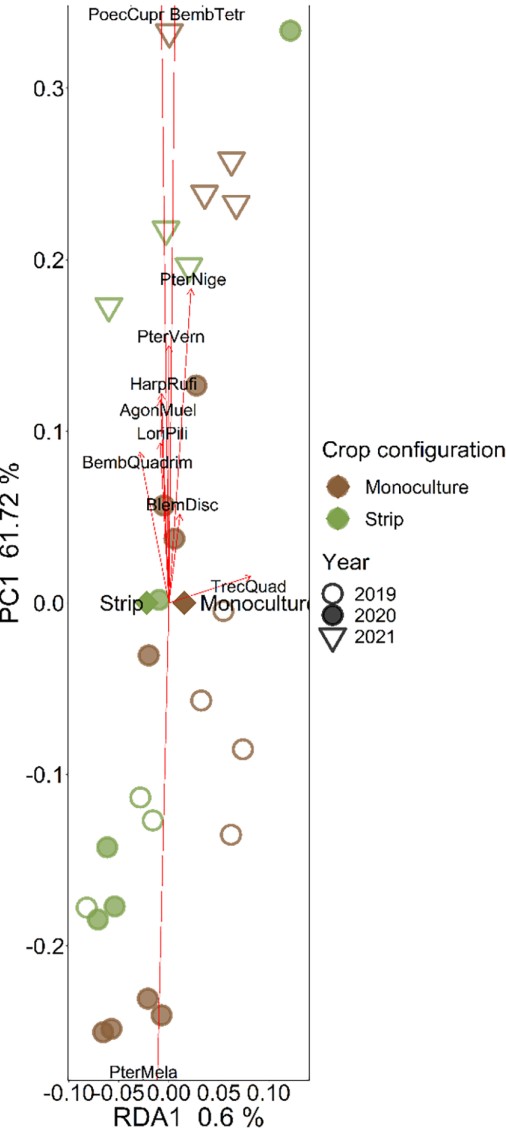

**Appendix 1—figure 3.** The effect of crop configuration on ground beetle community composition in Lelystad, using redundancy analysis (RDA). The first RDA and PC axis are visualized. Each axis shows the percentage explained variation. Shape of the dots indicates year (O=2019, ●=2020, ▽=2021), color of the dots indicates crop configuration (brown = monoculture, green = strip cropping), and the direction of each crop configuration on the RDA axis (x-axis) is mentioned. Red arrows indicate species placement and name codes are given close to the tip of the arrow, meaning of the name codes can be found in *Supplementary file 1*.

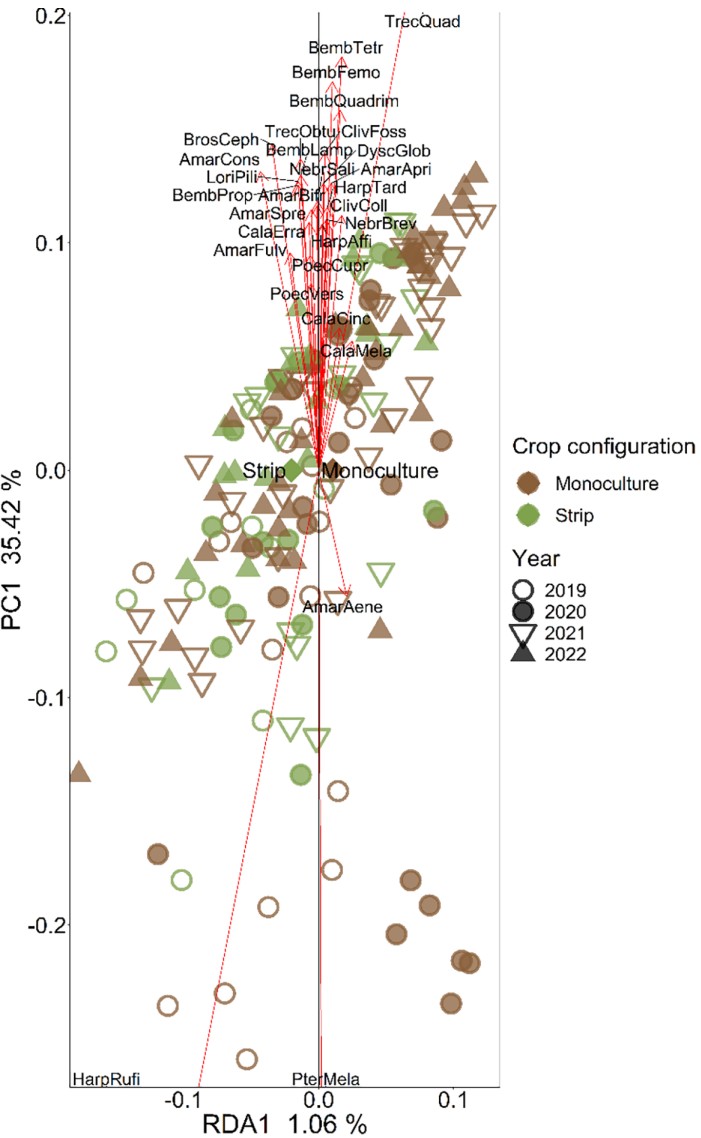

**Appendix 1—figure 4.** The effect of crop configuration on ground beetle community composition in Wageningen, using redundancy analysis (RDA). The first RDA and PC axis are visualized. Each axis shows the percentage explained variation. Shape of the dots indicates year (O=2019, ●=2020, ▽=2021, ▲=2022), color of the dots indicates crop configuration (brown = monoculture, green = strip cropping), and the direction of each crop configuration on the RDA axis (x-axis) is mentioned. Red arrows indicate species placement and name codes are given close to the tip of the arrow, meaning of the name codes can be found in *Supplementary file 1*.

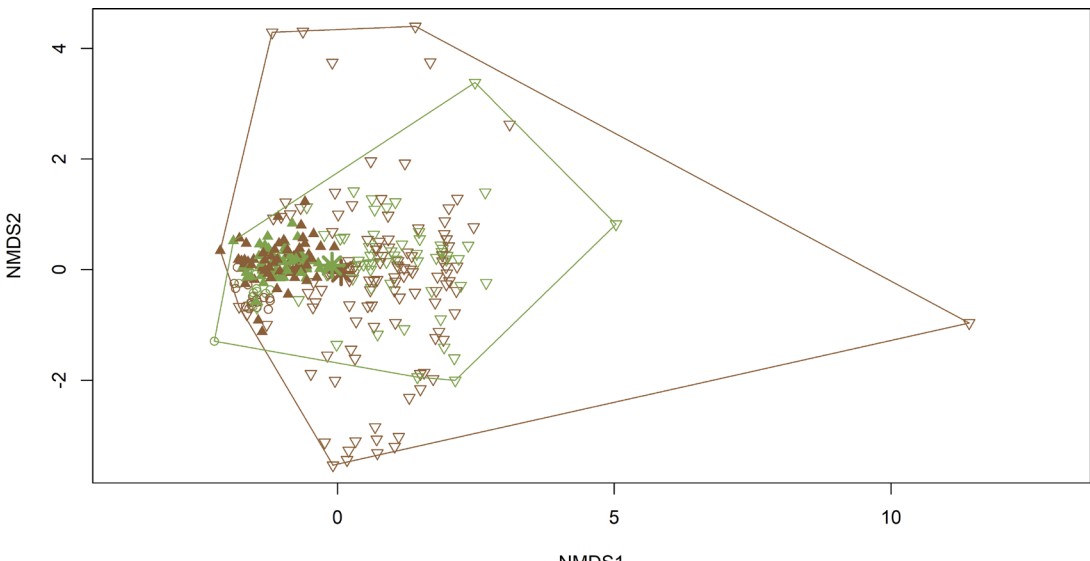

**Appendix 1—figure 5.** The effect of crop configuration on ground beetle community composition, using non-metric multidimensional scaling (NMDS). Data from all datasets that included species-level data (Almere [O], Lelystad [▽], Wageningen [▲]) are used for visualizing the two NMDS axes. Color of the dots indicates crop configuration (brown = monoculture, green = strip cropping), and the centroids of each crop configuration are indicated with a large asterisk (*).

*Appendix 1—figure 6 continued on next page*

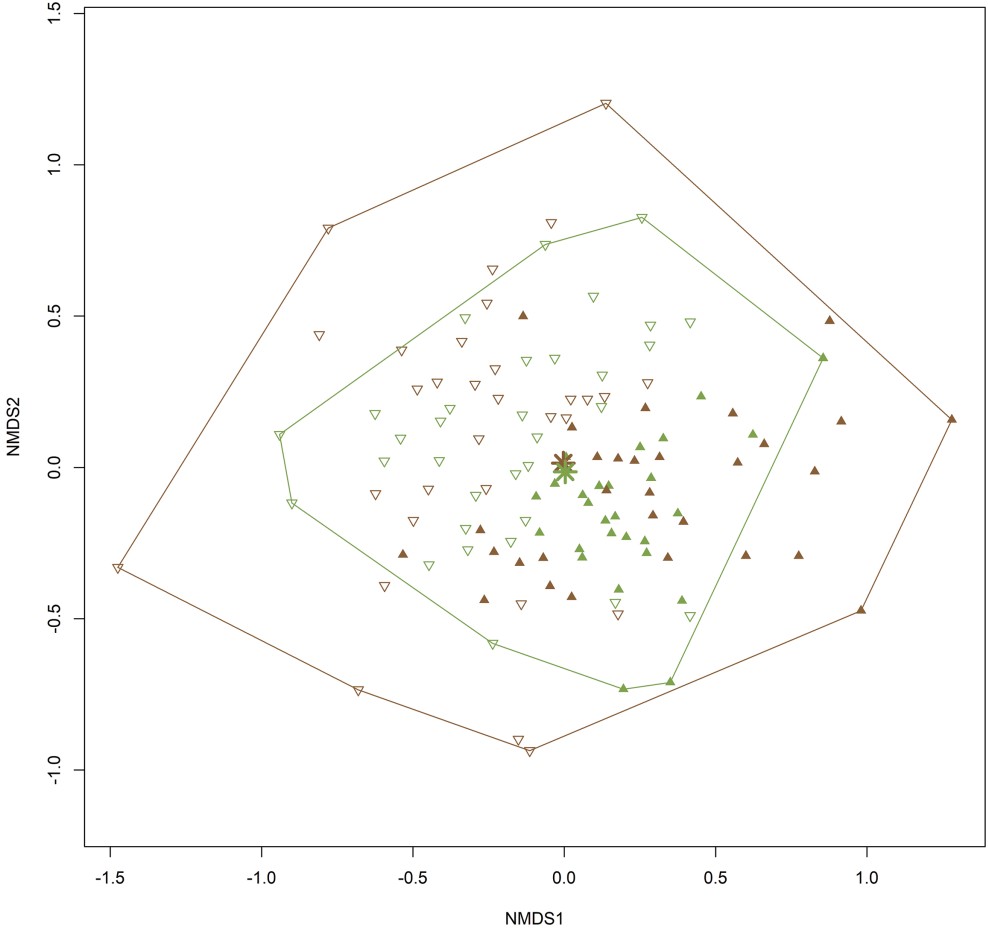

**Appendix 1—figure 6.** The effect of crop configuration on ground beetle community composition in Almere, using non-metric multidimensional scaling (NMDS). The two NMDS axes are visualized. Shape of the points indicates year (▽=2021, ▲=2022), color of the dots indicates crop configuration (brown = monoculture, green = strip cropping), and the centroids of each crop configuration are indicated with a large asterisk (*).

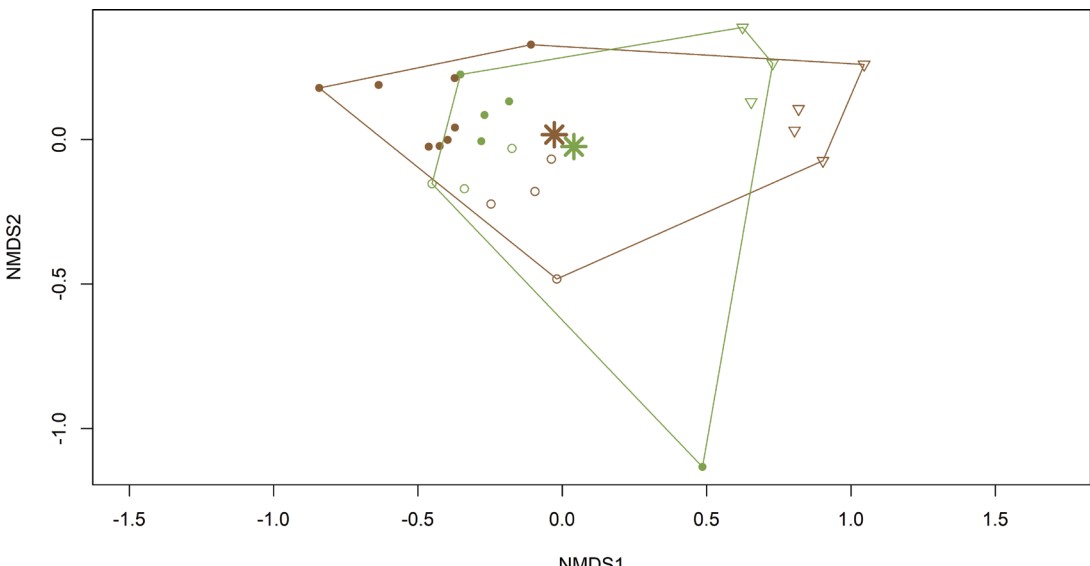

**Appendix 1—figure 7.** The effect of crop configuration on ground beetle community composition in Lelystad, using non-metric multidimensional scaling (NMDS). The two NMDS axes are visualized. Shape of
*Appendix 1—figure 7 continued on next page*

*Appendix 1—figure 7 continued*
the points indicates year (○=2019, ●=2020, ▽=2021), color of the dots indicates crop configuration (brown = monoculture, green = strip cropping), and the centroids of each crop configuration are indicated with a large asterisk (*).

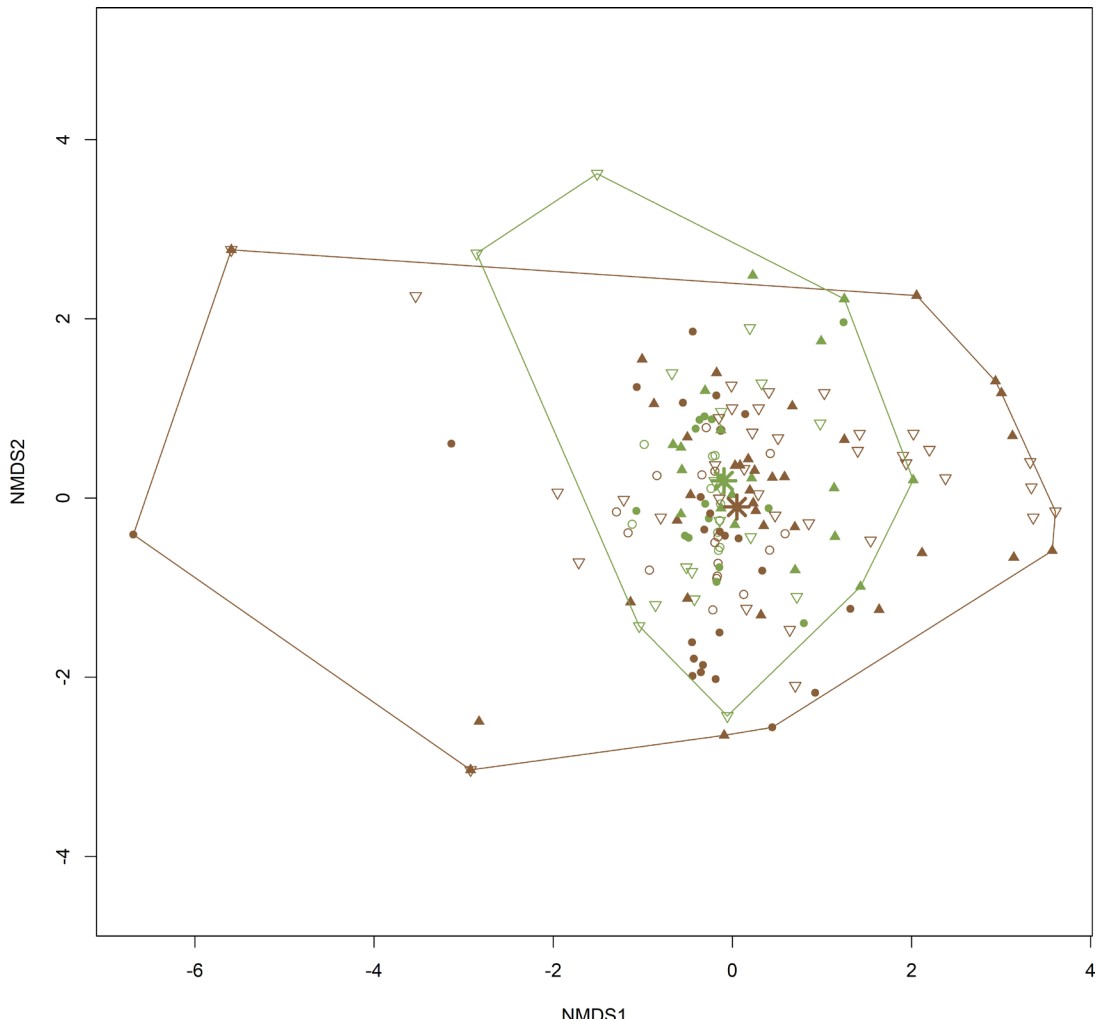

**Appendix 1—figure 8.** The effect of crop configuration on ground beetle community composition in Wageningen, using non-metric multidimensional scaling (NMDS). The two NMDS axes are visualized. Shape of the points indicates year (○=2019, ●=2020, ▽=2021, ▲=2022), color of the dots indicates crop configuration (brown = monoculture, green = strip cropping), and the centroids of each crop configuration are indicated with a large asterisk (*).

## Appendix 2

### Context dependency of crop configuration effects on ground beetle community composition. Case study in Wageningen

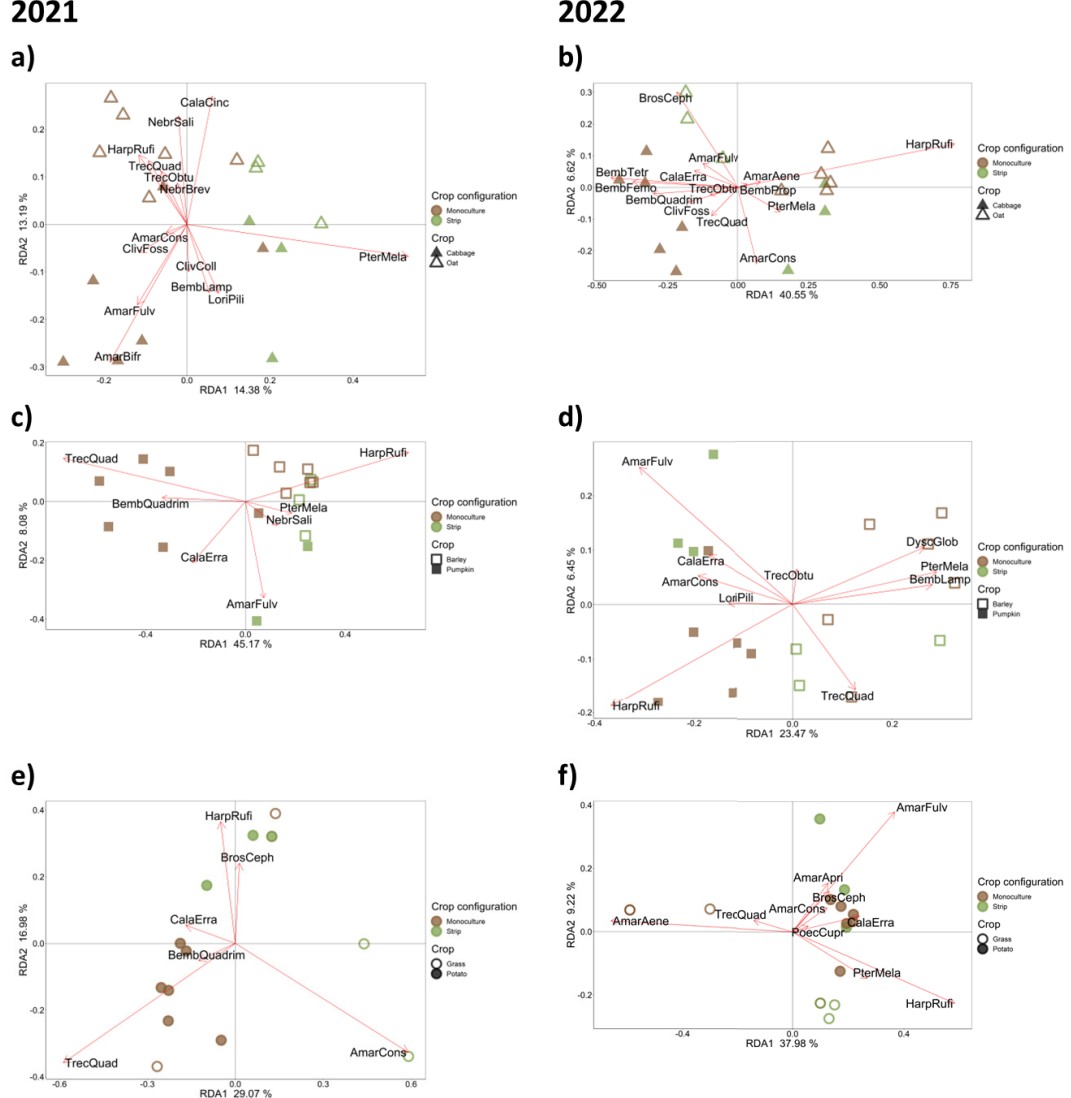

**Appendix 2—figure 1.** Effect of crop configuration and crop on ground beetle community composition in Wageningen. Here, we use data from Wageningen in 2021 (left column; **a**, **c**, **e**) and 2022 (right column; **b**, **d**, **f**) from three fields including three crop pairs being cabbage and oat (**a, b**), barley and pumpkin (**c, d**), and grass and potato (**e, f**). The first and second RDA axis are visualized. Each axis shows the percentage explained variation. Color of the dots indicates crop configuration (brown = monoculture, green = strip cropping), shapes indicate crop (□=barley, ■=pumpkin, O=grass, ●=potato, △=oat, ▲=cabbage). The red lines and abbreviated names indicate how specific ground beetles species correlate with the RDA axes, species names are given in **Supplementary file 1**.

