## [Editor Report · eLife Assessment]

This study presents **important** findings on increased ground beetle diversity in strip cropping compared with crop monocultures. **Solid** methods are used to analyze data from multiple sites with heterogeneous systems of mixed crops, allowing broad conclusions, albeit at the expense of lacking taxonomic specificity. The work will be of interest to all those applying plant diversity treatments to improve the diversity of associated animals in agricultural fields.

---

## [Referee Report · Joint Public Review]

Summary:

In this paper the authors examined the effects of strip cropping, a relatively new agricultural technique of alternating crops in small strips of several meters wide, on ground beetle diversity. The results show an increase in species diversity (i.e. abundance and species richness) of the ground beetle communities compared to monocultures.

Strengths:

The article is well written; it has an easily readable tone of voice without too much jargon or overly complicated sentence structure. Moreover, as far as reviewing the models in depth without raw data and R scripts allows, the statistical work done by the authors looks good. They have well thought out how to handle heterogenous, unbalanced and taxonomically unspecific yet spatially and temporarily correlated field data. The models applied and the model checks performed are appropriate for the data at hand. Combining RDA and PCA axes together is a nice touch. Moreover, after the first round of reviews, the authors have done a great job at rewriting the paper to make it less overstated, more relevant to the data at hand and more solid in the findings. Many of the weaknesses noted in the first review have been dealt with. The overall structure of the paper is good, with a clear introduction, hypotheses, results section and discussion.

---

## [Author Response]

The following is the authors’ response to the previous reviews

**Reviewer #3 (Public Review)**
Summary:In this paper the authors examined the effects of strip cropping, a relatively new agricultural technique of alternating crops in small strips of several meters wide, on ground beetle diversity. The results show an increase in species diversity (i.e. abundance and species richness) of the ground beetle communities compared to monocultures.Strengths:The article is well written; it has an easily readable tone of voice without too much jargon or overly complicated sentence structure. Moreover, as far as reviewing the models in depth without raw data and R scripts allows, the statistical work done by the authors looks good. They have well thought out how to handle heterogenous, unbalanced and taxonomically unspecific yet spatially and temporarily correlated field data. The models applied and the model checks performed are appropriate for the data at hand. Combining RDA and PCA axes together is a nice touch. Moreover, after the first round of reviews, the authors have done a great job at rewriting the paper to make it less overstated, more relevant to the data at hand and more solid in the findings. Many of the weaknesses noted in the first review have been dealt with. The overall structure of the paper is good, with a clear introduction, hypotheses, results section and discussion.

We are grateful for this positive feedback. We are glad that our extensive revision after extensive review from three reviewers has paid off in addressing earlier weakness of our manuscript.

Weaknesses:The weaknesses that remain are mainly due to a difficult dataset and choices that could have stressed certain aspects more, like the relationship between strip cropping and intercropping. The mechanistic understanding of strip cropping is what is at stake here. Does strip cropping behave similar to intercropping, a technique which has been proven to be beneficial to biodiversity because of added effects due to increased resource efficiency and greater plant species richness.Unfortunately, the authors do not go into this in the introduction or otherwise and simply state that they consider strip cropping a form of intercropping.

We agree with the reviewer that a mechanistic understanding on how intercropping and strip cropping differ would be very interesting. However, we also feel that this topic is somewhat beyond the scope of the current manuscript. We are already planning work to elucidate mechanisms that may explain the pest and suppressive effects of strip cropping.

I also do not like the exclusive focus on percentages, as these are dimensionless. I think more could have been done to show underlying structure in the data, even after rarefaction.

While we generally agree with this point raised by the reviewer, for our heterogeneous dataset it was difficult to come up with meaningful units with dimensions. Therefore, we believe that percentages are the most suitable approach to present readers a fair comparison of the treatments.

A further weakness is a limited embedding into the larger scientific discourses other than providing references. But this may be a matter of style and/or taste

We believe our manuscript to be well-embedded within the relevant scientific discourse, but as indicated by reviewer 3 this might indeed be a matter of style/taste. Without exact examples it is difficult for us to judge this point.

**Reviewer #3 (Recommendations for the authors):**
Suggestion for title: "Strip cropping shows promising preliminary increases in ground beetle community diversity compared to monocultures"

We agree that the title could indeed be nuanced. We incorporated the suggested title, except for the word “preliminary”, as we felt that this is slightly misplaced for a 4-year study conducted at 4 locations.

line 26: the word previous may be confusing to readers, as it suggests previous research on beetles or insects. I think it would be better to use for instance "related" or "productivity focused research"

We agree that this wording might be confusing, and changed it to “other studies showed”.

Line 84-85: this is vague. can you make explicit what you are trying to answer here?

We made “biodiversity metric changes” more explicit, and changed the sentence accordingly.

Line 88-89: I think this would fit better with the first question in line 83-84, so I suggest placing it upwards. Also, I think you mean abundant instead of common. Common suggests commonness in the entire population. Abundant suggests found often in this study. While these definitions may very much overlap, they are distinctly different.

We have moved this sentence up and changed “common” to “abundant”. To make the result section more in line with this section, we also moved the section on the relationship between crop configuration and abundant genera up.

Line 146: defining rareness of species should be in the methods section. Also "following" would be better than "according"

We now added a sentence on how we examine habitat preferences and rarity in the methods section (line 316-317). We also changed “according to” to “following”.

Line 291: it is called being "flush" with the soil surface. This expression is not much used by non-native speakers, but is regularly encountered in studies on pitfalls, so the authors could decide to change the sentence using the proper English vernacular.

Suggestion incorporated.

Line 322-327, this method could do with a reference

This method is a relatively standard calculation to calculate relative changes and to center variation around zero. Nevertheless, we added a reference to a paper that used the same method.

Line: 333-335. I would still like to see a reference for this method.

This methodology has not been described in literature to the best of our knowledge. As we compared two crops within strip cropping with their respective monoculture references, we compare one strip cropping field with two monocultural fields. Here we took a conservative approach by comparing the strip crop field with the monoculture with the highest richness and activity density, to see if strip cropped fields outperformed monocultures with diverse ground beetle communities.

Line 364-366. references?

We have added references for these R packages.